# Parametric design for soil gas flux system: a low-cost solution for continuous monitoring

Alex Naoki Asato Kobayashi[1], Clément Roques[1], Daniel Hunkeler[1], Edward A. D. Mitchell[2], Robin Calisti[2], and Philip Brunner[1]

[1]Centre d'hydrogéologie et de géothermie (CHYN), Université de Neuchâtel
[2]Laboratory of Soil Biodiversity, Université de Neuchâtel
[1,2]Rue Emile-Argand 11, Neuchâtel, Switzerland

**Correspondence:** Alex Naoki Asato Kobayashi (alex.asato@unine.ch)

**Abstract.**

Monitoring soil gas fluxes is essential for understanding greenhouse gas dynamics within the critical zone. One commonly used method involves chamber-based methods which enables the quantification of soil gas fluxes on a specific point at the soil-atmosphere interface. However, point measurements often limit the representativeness of the field-scale processes due to the large spatio-temporal variability of climatic, hydrological, pedological, or ecological factors controlling its dynamics. Additionally, commercial chambers often prohibit deployment over sufficient representative area due to expensive operational and purchase costs.

Although low-cost and open-source designs have recently emerged in the literature, solutions enabling adaptability to field-site characteristics and design validation are still lacking. To address these challenges, we propose here a low-cost, parametric soil gas flux system that can be adapted to logistical and field constraints while allowing high-frequency measurement resolution. Along with open-design hardware, we developed software to automate data collection and processing. Laboratory tests, including static and transient experiments, assessed both the sensor and chamber design integration. Our results show strong agreement between the low-cost and commercial gas analyzers in static conditions with $CO_2$ levels up to 500 ppm above background. During transient tests, we successfully replicated $CO_2$ concentration increases using both systems, with comparable response time between eeflux and measurement from the sensors. Thus, we were able to address the data reliability from the low-cost setup, despite different parameters. Ultimately, our approach demonstrates that low-cost solutions can democratize these systems through a flexible framework suitable for various study sites.

## 1 Introduction

Anthropogenic emissions and removal of greenhouse gases (GHG) in agricultural, forestry, and other land uses are driven by biological and physical processes, which can be determined by ecosystem components such as biomass, dead organic matter, soils, and livestock (IPCC, 2006, 2019, 2021). Most forms of GHG emission through the soils are via microorganisms, root respiration, and chemical processes. Several environmental factors influence the dynamic of $CO_2$ fluxes, methanogenesis and

nitrification, and denitrification, with the main ones being temperature, soil water content, nutrient availability, and pH (Oertel et al., 2016; Smith et al., 2018).

Several soil gas flux monitoring technologies exist, with two main approaches being widely used in environmental science: the Eddy Covariance method (EC) and the chamber-based method. The Eddy Covariance method uses a high-frequency anemometer and gas analyzer (around 20 Hz) to calculate the net flux for water vapor or $CO_2$, $CH_4$ or $N_2O$ (Aubinet et al., 2012). From the turbulent winds, this direct method measures the covariance from the eddies and the trace gases it carries. Despite this method contributing largely to the understanding of the global change biology (Baldocchi, 2020; Baldocchi et al.,
2001), it is not a substitute for the use of chamber-based methods because it overlooks small-scale variability of trace gas fluxes (Eugster and Merbold, 2015; Baldocchi, 2014). Furthermore special care must be taken regarding flux footprints in highly heterogeneous landscapes (Chu et al., 2021), despite continuous measurements from EC, significant uncertainties remain in measuring the net exchange of GHG (Eugster and Merbold, 2015).

    Meanwhile, chamber-based methods are essential to understanding soil gas flux point variability that is affected by topogra-
phy, vegetation, soil types, and hydrology. These chambers can be static, dynamic, manually, or semi-autonomously operated. While the manual chamber approach has been widely used because it only requires one set of chamber and gas analyzer, it is also not capable to performing measurements without an operator, thus it can be hard logistically to have high temporal resolutions with this approach. In contrast, semi-autonomous systems are particular interesting because it can achieve a high temporal resolution in less accessible places. However, the chamber's reduced scale compared to EC comes with several chal-
lenges regarding possible measurement artifacts. Maier et al. (2022) aggregated the experience of different groups to define some "best practices" when measuring soil gas flux. The disposition of chambers in a study site is critical and requires in depth understanding of the ecosystem and a clear definition of the goal of a project, e.g. the goal can be to test the effect of a field manipulation on an "average" representative microsite replicated over the field site. Depending on how it is performed, there is the potential of underrepresenting the spatial variability in ecosystems with hotspots or high flux events (Leon et al., 2014;
Jenerette et al., 2008), especially when the cost of the commercially available chambers and gas analyzers is often prohibitively expensive for some project budgets to distribute several chambers throughout a study site properly.

    Advancements in low-cost sensors have improved their affordability and accuracy, creating new opportunities to explore these alternatives for measuring soil gas flux. Some alternatives of manual respiration chambers (Zawilski and Bustillo, 2024; MacAgga et al., 2024) or more autonomous systems (Gagnon et al., 2016; Forbes et al., 2023) have been developed to meet the
demand for near-continuous monitoring. Their validation is often made using only the sensor's static accuracy. i.e measuring a given concentration of a standard gas, or duplicate the measurements in the field with a commercial sensor installed in parallel. However, in this case it remains challenging to compare the results due to bias in the sampling footprint. Additionally, it is well-established that chamber and system design can significantly reduce flux errors when appropriately tailored to the physical characteristics of a given soil (Venterea and Baker, 2008). We identified a lack of parametrized designs that allow the chambers
to be produced to fit each study site's conditions and a lack of more reproducible laboratory methodologies that try to simulate soil gas flux signals regarding the chamber methods constraints.

Our main objective is to present a low-cost sensor system for soil gas flux, along with a parametrized chamber design that can be adapted to the specific conditions of any study site. Additionally, we propose a laboratory methodology that allows users to test their designs by implementing a transient experiment. This approach enables users to evaluate the dynamic response time of $dCO_2/dt$ in a controlled environment, ensuring the reliability and accuracy of soil gas flux systems before field deployment.

## 2 Material and Methods

This study aimed to increase the spatiotemporal representativeness of soil gas flux measurements, enabling the capture near-continuous soil respiration and net ecosystem exchange (NEE) at a given study site. To achieve this, we developed an open-design, low-cost parametric soil gas flux system and proposed a novel laboratory methodology for testing its performance.

### 2.1 Soil gas flux sensor system: Open-design

The designed soil gas flux sensor system consists of (i) a parametric soil gas flux chamber, (ii) integrated sensor components, and (iii) an integrated controller PCB (Figure 1).

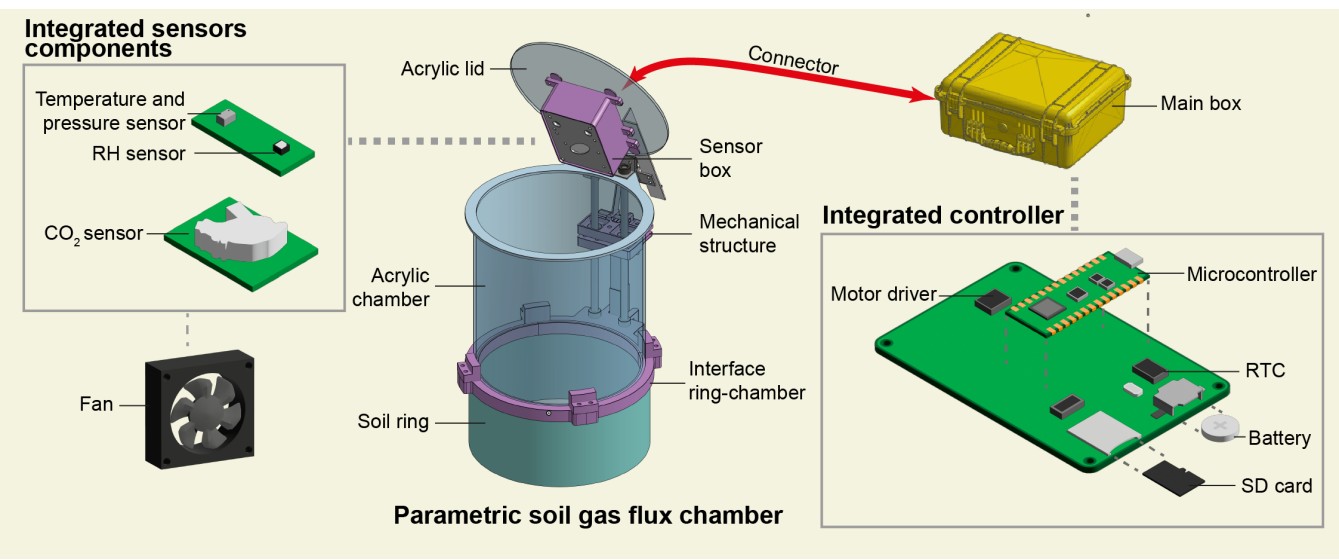

**Figure 1.** Schematic representation of low-cost $CO_2$ soil gas flux system indicating the chamber components with sensor box PCB and fan components and main box with controller PCB.

### 2.1.1 Parametric soil gas flux chamber

The main design philosophy for the soil gas flux chamber was the parametrization of the chamber's dimensions. Parametrizing the chamber dimensions is crucial for accurately calculating soil gas flux, as different volume-to-surface area ratios significantly influence the evolution of $CO_2$ concentration within the chamber over time. Some additional implications from the change in

chamber dimensions are the air mixing in the chamber when measuring, the optimal range of $CO_2$ concentration inside the chamber, the minimal detectable flux (section 2.1.3) that the system can calculate, and the area-perimeter ratio to increase the representativeness for a given measurement point.

Thus, two critical parameters can be adapted: chamber diameter and height. These variables directly affect the chamber's volume and influence the rate of gas accumulation during measurements. The diameter determines the area over which gas is sampled. The parametrization of the chamber is field-specific, starting with the expected range of soil gas flux and sensor characteristics, which are further constrained by vegetation and material availability (Figure 2).

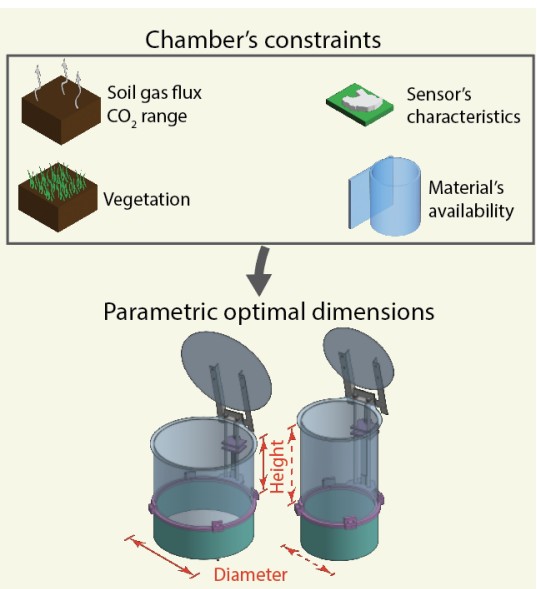

**Figure 2.** Parametrized chamber design based on chamber's constrains such as soil gas flux range, sensor characteristics, vegetation type and user's material availability

### 2.1.2  Electronics: sensors and microcontroller

The electronics are divided into two parts: (i) integrated sensor components and (ii) integrated controller PCB. The integrated sensor components is located in the soil gas flux chamber, where the measurement is happening, while the main section is located inside an encasing with the battery and any other main section from other chambers (Figure 1). Each section is in its weather-protected box, and the microcontroller communicates with the sensor section via $I^2C$ protocol. This communication protocol is digital, and it uses a serial data line (SDA) and a serial clock line (SCL).

Since our objective is to have a system suitable for continuous measurements, the microcontroller has a sequence of events divided into four phases: (i) closing chamber, (ii) measurement, (iii) opening chamber, and (iv) standby. The closing chamber phase activates the motor driver to close the linear actuator and activates all sensors to start their measurement. After the chamber is closed, the measurement phase starts logging the data from all sensors and the fan begins to mix the air inside the

chamber. All data from the measured sensors are logged using the micro-SD module, which also keeps the clock time using the real-time clock module (RTC). After the configured measurement period is finished, the opening chamber phase stops the measurements from the sensors and fan, and then the chamber is opened. Finally, the standby phase deactivates all sensors and waits for the user-defined waiting period to finish.

For debugging purposes, we configured the microcontroller to identify, log, and restart the system to its original settings in case of unexpected error or shutdown. This ensures that the user can identify the cause of an error and fix it for future operations.

All sensors and main components are found at the table 1. In order to estimate soil gas flux, it is necessary the $CO_2$ gas concentration, atmospheric pressure, temperature and humidity sensor. Among those sensors, the cost magnitude difference from the commercial sensor can reach a thousand fold higher compared with the low-cost sensors system. Also, the PCB design and schematics are found at the repository (Kobayashi, 2025).

| Component | Manufacturer | Description |
|---|---|---|
| K30 FR | Senseair | $CO_2$ concentration sensor |
| BMP280 | BOSCH | Atmospheric pressure and temperature sensor |
| SI7021 | Silicon Laboratories | Humidity and temperature sensor |
| Raspberry Pi Pico W | Raspberry Pi Foundation | Microcontroller |
| TB6612FNG | Toshiba Semiconductor | Motor driver |
| DS1307 | Dallas Semiconductor | RTC |
| CD74HC4050 | Texas Instruments | SD driver |

**Table 1.** List of electronics components

### 2.1.3 Minimal detectable flux

After considering the chamber design and sensors, we can start integrating both, and the minimal detectable flux, which was proposed by Christiansen et al. (2015), provides what is the expected lower limit for the soil gas flux rates that can be detected. The minimal detectable flux (MDF) considers different characteristics, such as the analytical accuracy of the sensor, the chamber's volume and surface area, and the total chamber closure time. Nickerson further developed equation 1 for instruments that have a sampling time with a higher frequency (e.g. 1 Hz).

$$\text{MDF} = \left( \frac{A_a}{t_c \sqrt{\left( \frac{t_c}{p_s} \right)}} \right) \left( \frac{VP}{SRT} \right) \tag{1}$$

Where $A_a$ $[ppb]$ is the analytical accuracy of the instrument, $t_c$ $[s]$ is the closure time of the chamber, $V$ $[m^3]$ is the chamber volume, $P$ $[Pa]$ is the atmospheric pressure, $S$ $[m^2]$ is the chamber surface area, $R$ $[m^3\ Pa\ K^{-1}mol^{-1}]$ is the ideal gas constant, $T$ $[K]$ is the ambient temperature, and $p_s[s]$ is the sampling periodicity .

## 2.2 Laboratory experiment

We performed two laboratory experiments: the first was designed to check the sensor accuracy by imposing a known $CO_2$ concentration, further mentioned as a static $CO_2$ concentration experiment (section 2.2.1). The second, aiming at evaluating the sensor's performance in measuring $CO_2$ changes over time in the chamber, which ultimately is used to estimate the corresponding flux, is further referred to as transient $CO_2$ experiment (section 2.2.2).

We used bags containing standard $CO_2$ concentrations for both the static and advection-driven simulated soil gas flux experiments. To prepare the standard bags, we used a 5 liters Tedlar sampling bag with dual stainless steel fittings, where we mixed a given volume of a 99.9% concentration of $CO_2$ gas with atmospheric air.

### 2.2.1 Static experiment

The static $CO_2$ concentration test checks the individual sensor's accuracy of the low-cost sensor against the commercial gas analyzer (LICOR LI-7810) without considering the overall chamber. To ensure both sensors are exposed to the same concentration of gases, the standard $CO_2$ gas was injected directly through the LI-7810 sensor and then routed to the inlet of the low-cost sensor (K30). In addition, we ensured that the internal clock of both systems was synchronized. Using this approach, we logged the data continuously while the standard $CO_2$ bag was connected to the inlet of the LI-7810 sensor. Thus, we logged 1502 points from the low-cost sensor and LI-7810 during 12 different standard $CO_2$ gas bag concentration, ranging from 200 ppm until 1100 ppm above the $CO_2$ background concentration.

### 2.2.2 Transient $CO_2$ experiment: dynamic response on the system

The transient $CO_2$ experiment aim to better understand the systems' temporal $CO_2$ concentration evolution inside an accumulation chamber given a known $CO_2$ standard gas concentration. The first step for this experiment was to measure the concentration of the imposed $CO_2$ gas bag standard. This was done by measuring with the LI-7810 the gas bag concentration. For this experiment, we targeted a concentration of 900ppm +/-10%, for which we had a total of 8 $CO_2$ gas bags. The second step consists of measuring the $CO_2$ accumulation in the commercial chamber (LICOR 8200-01S) with the gas analyzer (LI-7810) and our design systems. This is done by closing the respective chambers and measuring the evolution of CO2 concentration until the gas bag is empty (Figure 3). Finally, we open the low-cost parametric chamber to return to the atmospheric background $CO_2$ concentration.

The evaluation procedure for both systems' transient $CO_2$ experiment utilizes a numerical model estimation and compares them with the measured data. The assumptions for this model are constant pressure, temperature, and humidity throughout the experiment. With constant pressure, a leakage term from both systems is expected and is considered for our transient $CO_2$ experiment. So, the influx equals the outflux (leakage term), or the total volume is conserved during the experiment period. From this transient experiment, we modeled the chamber $CO_2$ mass, thus following the equation 2.

$$\text{Chamber Mass}_{CO_2}(t) = \text{Chamber Mass}_{CO_2}(t=0) + \dot{m}_{CO_2} - \dot{o}_{CO_2} \tag{2}$$

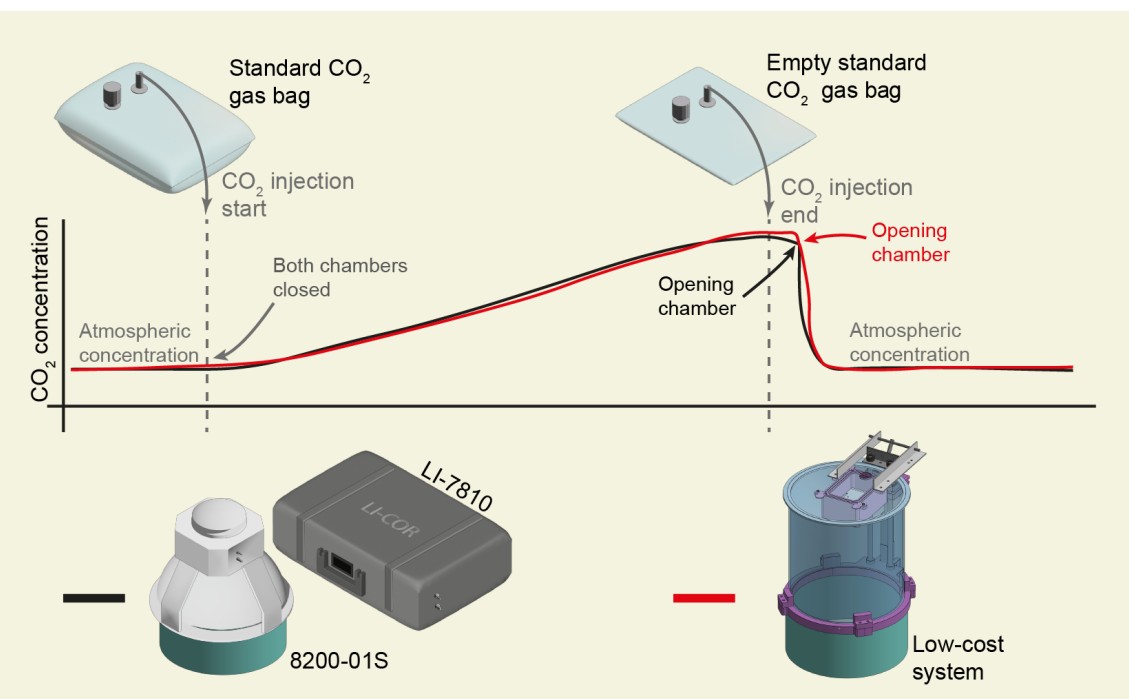

**Figure 3.** Schematic laboratory experiment using the transient $CO_2$ experiment, from background atmospheric $CO_2$ concentration to the increase of concentration after the chamber is closed then the opening of chamber back to the atmosphere and an additional step for chamber leak for the low-cost system. The black line represents our reference (8200-01S and LI-7810) and the red line represents our low-cost system.

Which, $\dot{m}_{CO_2}$ is the standard $CO_2$ mass gas input to the chamber and $\dot{o}_{CO_2}$ is the leakage of $CO_2$ from the chamber and Chamber Mass$_{CO_2}$ is the $CO_2$ mass in the chamber.

Through modeling, we can compare the expected $CO_2$ curve evolution through time with the measured data, but to compare the system between each other, it is necessary to perform an adimensional analysis. There are differences between our reference (LI-7810 and 8200-01S) and low-cost sensor systems regarding the setup, such as chamber volume, initial $CO_2$ concentration, standard $CO_2$ concentration gas bag, and injection flux rate. And to compare both system that utilized different setups, we normalized the time (Equation 3) and the concentration (Equation 4).

$$\hat{t} = \frac{t}{\frac{V_{chamber}}{Q}} \tag{3}$$

Where, $\hat{t}$ is the dimensionless time, $t$ is time, $V_{chamber}$ is the chamber volume and $Q$ is the standard $CO_2$ gas bag of $CO_2$ injection flux rate.

$$\hat{C} = \frac{C - C_0}{C_{inj} - C_0} \tag{4}$$

Where, $\hat{C}$ is the dimensionless $CO_2$ concentration, $C$ is the $CO_2$ concentration at a given time, $C_0$ is the initial $CO_2$ concentration before the injection start and $C_{inj}$ is the $CO_2$ concentration from the standard gas bag.

## 2.3 Field experiment

To evaluate the reliability and performance of our newly developed system under real-world conditions, we deployed it in the field, focusing on its ability to capture soil gas flux dynamics accurately and consistently in a natural setting. The experimental site is located on the elevated Swiss Plateau in the city of Kerzers, in the canton of Fribourg, Switzerland. The region's climate is temperate, with annual precipitation between 800 and 1400 mm and an annual temperature of 8 to $12°C$ and the site's elevation is approximately 432 m.a.s.l. and this regions is considered the most productive region in Switzerland for
vegetable production. However, the agriculture situation in this region has become critical because of its highly organic soil type characteristics (peaty soil). And with the drainage requirements for agriculture, the carbon storage has been depleted by the exposition from the drainage areas, causing subsidence and significant emissions of $CO_2$ (Egli et al., 2021; Roeoesli and Egli, 2024). Thus, to promote sustainable management of these cultivated peaty soils, monitoring those emissions is required to evaluate the efficacy of the management of these soils (Ferré et al., 2019).
There are two main calculation models used for estimate soil gas flux is the linear and the non-linear model. We calculated the $CO_2$ soil gas flux using the H-M model (Pedersen et al., 2010), which uses an exponential equation (equation 5) to fit the $CO_2$ concentration data and it is considered more accurate than the linear model (Baneschi et al., 2023). After fitting the measured data to the equation, we can extract the fitted variables and then calculate the $\frac{dC}{dt}$ (equation 6).

$$C_c(t) = C_x + (C_0 - C_x)e^{-\alpha(t-t_0)} \tag{5}$$

where $C_0$ $[ppm]$ is the initial concentration of $CO_2$ before the chamber was closed, $C_x$ $[ppm]$ is the soil gas concentration of $CO_2$ in the soil, $\alpha$ $[m\ s^{-1}]$ is the $CO_2$ soil gas conductivity, $C_c$ $[ppm]$ in the function of time is the chamber concentration, and $t$ $[s]$ is time since the chamber is closed.

$$\frac{dC}{dt} = \alpha(C_x - C_0)e^{-\alpha(t-t_0)} \tag{6}$$

And finally, the soil gas flux can be calculated using the following equation 7.

$$F_{CO_2} = \frac{10VP_0(1-W_0/1000)\frac{dC}{dt}}{RA(T_0+273.15)} \tag{7}$$

$V$ $[cm^{-3}]$ is the chamber volume, $A$ $[cm^{-2}]$ is the area of the corresponding chamber ring, $P_0$ $[kPa]$ is the initial pressure of the chamber, $W_0$ $[mmol\ mol^{-1}]$ is the water humidity vapor, and $T_0$ $[°C]$ is the initial chamber temperature.

Past field campaigns gave us some insights about the minimal soil respiration for this study site of around $2000\ mmols\ m^{-2}\ s^{-1}$, our low-cost soil gas flux sensor system setup consists of continuous measurements of 200 seconds, with a total of 4 repeti-

tions and a standby time of 25 minutes. The goal was to assess the in-situ capacity of continuous measurement in the field environment while measuring the diurnal cycle of the $CO_2$ soil respiration.

## 3   Results

### 3.1   Parametric chamber design

The open-source parametric chamber design, integrated sensor PCB, integrated controller PCB, and software can be found
in the public repository (Kobayashi, 2025). In this repository, we split into chamber parametric design for the assembly and software for control and operation of the whole system. The parametric design starts when the user has to define an initial expected soil gas flux $CO_2$ range to optimize the chamber's dimension for this minimal soil gas flux, and to calculate the expected range of concentration of $CO_2$ for any given measurement time. Additionally, we take the sensor's manufacturer accuracy, the list of possible materials from the user, the vegetation's expected height if measuring the net ecosystem exchange
(NEE), and the user's measuring time to consider the minimal dimensions for the chamber. This information from the user is then utilized to define a list of options that best fit the user's study site based on the minimal detectable flux, try to maximize the base area-perimeter ratio (Maier et al., 2022) while trying to maintain the chamber's concentration optimal range of $CO_2$ concentration.

For our choice in the design for our specific study site, the low-cost $CO_2$ sensor has an accuracy of 30 ppm, and we initially
defined a measurement time of 150 seconds which a chamber height of 50 cm would get us a minimal detectable flux of $350\ nmols\ m^{-2}\ s^{-1}$ (Figure 4). Inversely, the maximum expected soil gas flux constrains the chamber height so that the $CO_2$ gas concentration inside the chamber stays less than $\Delta$ 500 ppm above the background concentration, thus we choose a height of 20 cm with the chamber diameter of 20 cm also to fit material availability constraints. With this diameter, the ratio area-perimeter is 5 cm, which is considered acceptable regarding the relative disturbance from the soil ring edge along the base
(Healy et al.; Maier et al., 2022). Also, this reduction in chamber height implied that we improved the minimal detectable flux to around $120\ nmols\ m^{-2}\ s^{-1}$ if we later decide to switch from soil respiration to NEE, or we also could reduce the measuring time and maintain a higher minimal detectable flux (Figure 4).

### 3.2   Laboratory analysis: static test

For the static $CO_2$ concentration test, we considered the relative difference between the background $CO_2$ concentration
recorded by each sensor and the concentration of $CO_2$ measured from the standard $CO_2$ bag, labeled $\Delta CO_2$. Our approach showed a good linear correlation between the two sensors ($y = 1.005x$) for $\Delta CO_2 < 600$ ppm measured with the LI-7810 (Figure 5). For $\Delta CO_2 > 600$ ppm, the regression slightly changed with lower values measured by our low-cost sensor with respect to the commercial LI-7810. Considering this higher concentration range, the linear regression tended to $y = 0.946x$ (Figure 5). Thus, the low-cost sensor underestimates in comparison to the LI-7810 when introducing a standard $CO_2$ gas
concentration of $> 600$ ppm above the background concentration.

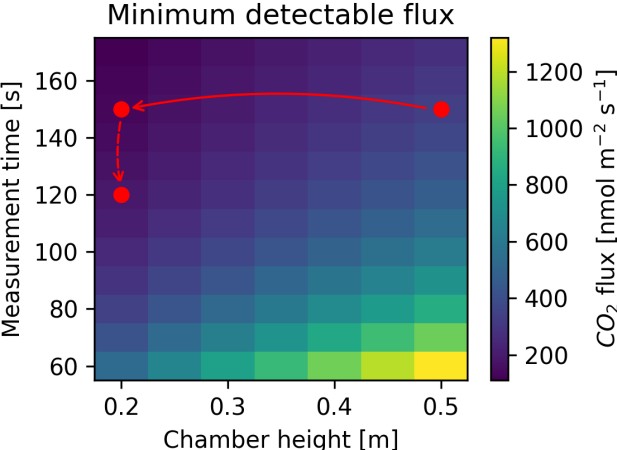

**Figure 4.** Matrix of minimum detectable flux given chamber height and measurement time, where we change our chamber design and measurement time to adjust the minimum detectable flux

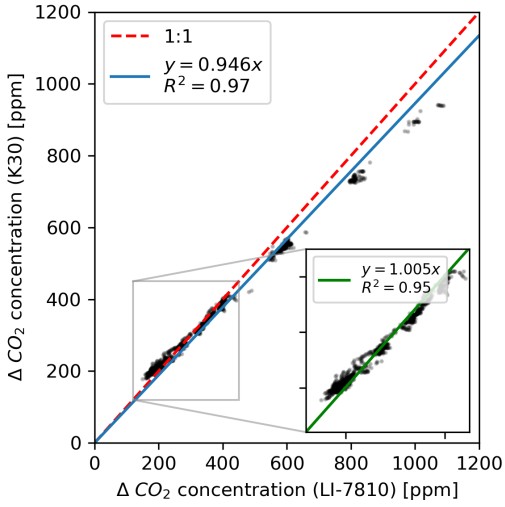

**Figure 5.** Static $CO_2$ concentration comparison between LI-7810 and low-cost sensor K30 relative to their background concentration, where it shows the linear relationship for the whole range ($y = 0.946x$) and zoom in to the range between 150 and 500 ppm ($y = 1.005x$).

### 3.3 Laboratory analysis: transient $CO_2$ experiment

Our results from the laboratory analysis based on the transient $CO_2$ experiment show how their absolute concentrations differ from the reference to the low-cost approach. The $\Delta CO_2$ concentration for the reference ranges from 0 ppm to 90 ppm, and for

the low-cost approach, the range is from 0 ppm to 190 ppm; in addition, the measuring time difference between the reference

is 120 seconds, while the low-cost sensor is about 180 seconds.

To allow us to compare two different setups (chamber volume and initial conditions), we utilized the adimensional concentration and time to normalize the inputs from each system. The normalization can be observed in Figure 6, which shows that the measured data fits closest with the modeled curve for both the reference and low-cost system. By checking their respective residuals, the measured data maintained stability over the entire measurement period after a slight increase at the

beginning. These results show that both systems were able to mix the air inside the chamber properly when measuring the $CO_2$ concentration and confirm the air tightness of the system that injects the standard gas.

The initial offset on the residuals (Figure 6) from both reference and low-cost systems indicated a response lag, which comes from the initial injection of the standard $CO_2$ gas and mixing time in the chamber. This offset from the reference was lower when compared to the low-cost sensor, indicating that the lag from the low-cost system was higher than our reference.

Converting back to time, the lag from the commercial system was about 5 seconds, while the low-cost system was about 20 seconds.

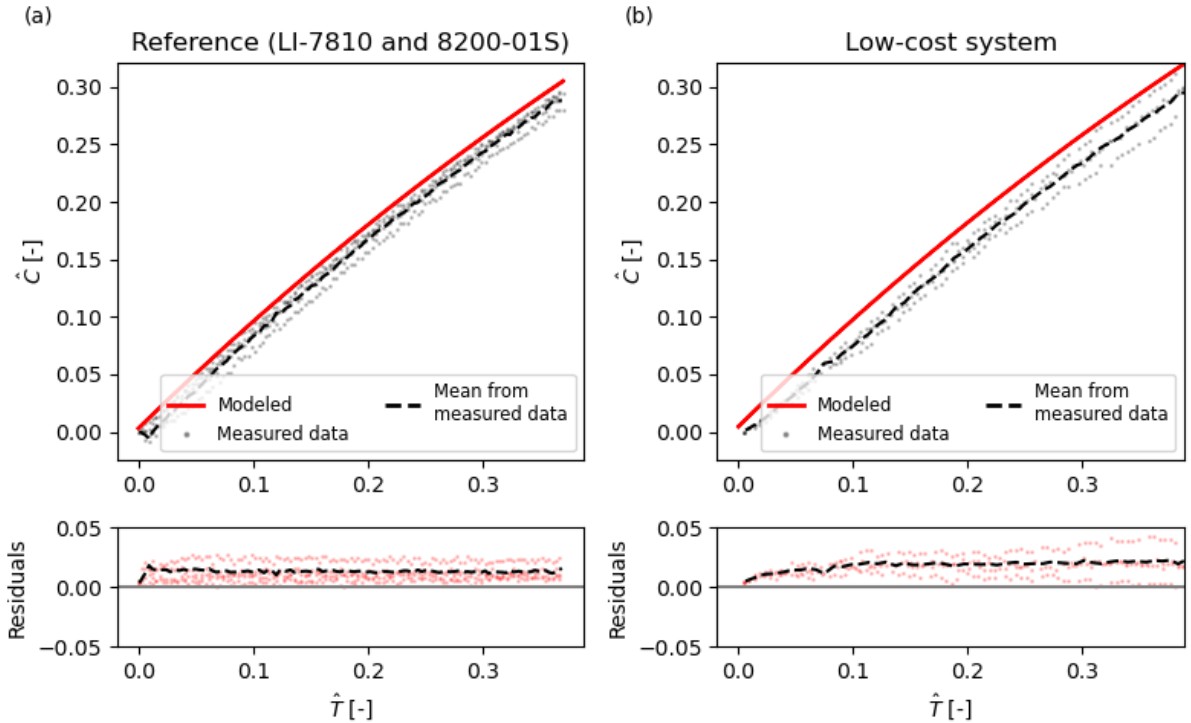

**Figure 6.** Dimensionless concentration plotted against dimensionless time, showing residuals between the measured concentration and the expected modeled curve. Panel (a) displays results from the LI-7810 sensor, while panel (b) shows results from the low-cost sensor system.

### 3.4 Field deployment

Our field experiment was performed from June 12th to 14th of 2024, and a total of 314 $CO_2$ soil gas flux measurements were made during this period. This brief snapshot of $CO_2$ concentration showed the diurnal cycle (Figure 7a). The initial concentration of $CO_2$ changed from a range of 430 ppm during the daytime to 600 to 800 ppm during the nighttime. For the $CO_2$ soil gas flux (Figure 7b), values of around 4.5 $\mu mol\ m^{-2}\ s^{-1}$ was measured with a peak of around $16:00$ for both days, but during the nighttime, the value fluctuated around 2 $\mu mol\ m^{-2}\ s^{-1}$ and with a variance higher (0.14) than the daytime (0.02). During this experiment there was no precipitation, thus we can see the slight decrease in top soil water content (Figure 7), where during the night the soil water content decrease is lower than compared during the day. Also, we can see the soil temperature diurnal cycle, which ranged from 15 $°C$ during the night to around 18 $°C$ at around $16:00$.

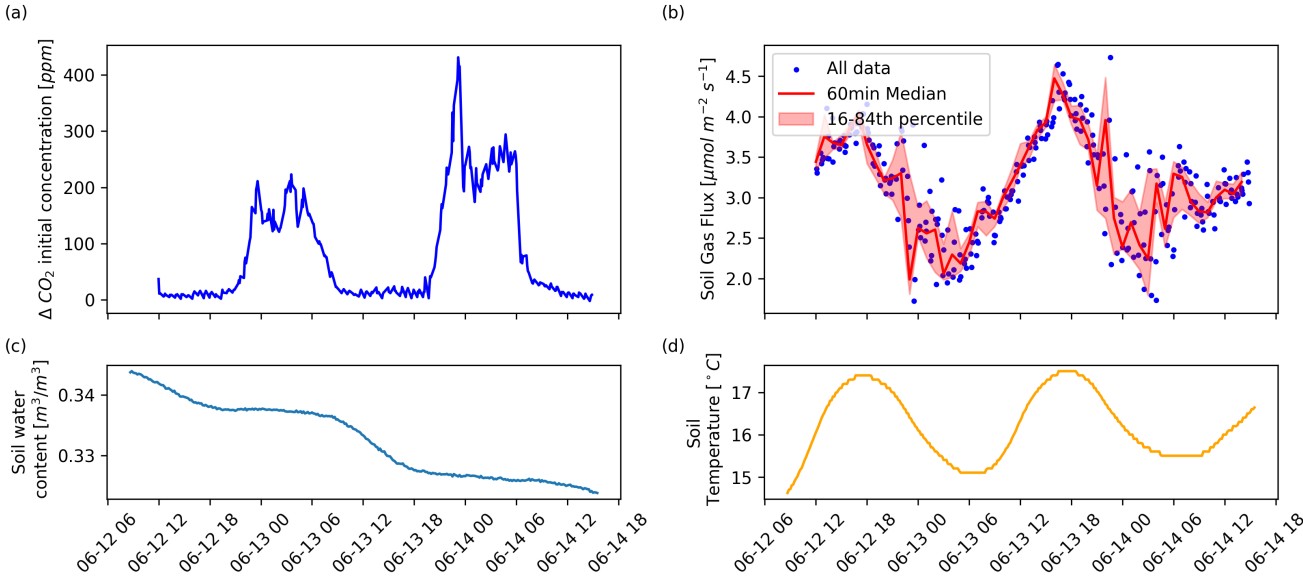

**Figure 7.** Field measurement of the low-cost soil gas flux system for two day period where it shows (a) initial concentration of $CO_2$ at each measurement, (b) the soil gas flux of $CO_2$ with median of the moving window of 60 minutes, (c) the top soil water content and (d) the top soil temperature.

## 4 Discussion

### 4.1 Overall challenges and opportunities in soil gas flux measurement

Measuring soil gas flux is inherently challenging due to the complex interactions between the soil, vegetation, and atmosphere. The combination of accumulation chambers and Eddy Covariance systems has proven valuable applications (Basri et al.,

2024; Alice Courtois et al., 2019), but the high costs of commercial dynamic chambers limit their scalability and accessibility for spatiotemporal variability analysis. In this context, low-cost sensor systems (Gagnon et al., 2016; Forbes et al., 2023; MacAgga et al., 2024) offer a promising alternative or complement, providing an affordable and scalable solution for soil gas flux monitoring.

Custom chambers have the flexibility to be parametrized to adapt their design to different study sites and expected flux 245 conditions beyond the specific conditions for which they were originally developed. However, their development represent an investment and various expertise are required for their initial design and testing. In addition, while custom chambers and low-cost sensors enable an increase in the number of measurement points, they also transfer the burden of ensuring data quality to the user, who must validate the performance of these newly designed low-cost systems.

These challenges shaped our study's objective: to develop a parametric and open-source design adaptable to diverse study site 250 conditions, maintain the low-cost system approach, and propose a transient $CO_2$ experiment method to evaluate the efficiency of custom chambers and ensure seamless integration between chamber and sensor. Such integrated solution from the chamber and sensor implies the necessity of having a more holistic approach that is capable of evaluating the performance of an entire system instead of a single component.

## 4.2 Performance validation: laboratory static and transient experiment and field application

For the parametric chamber design and overall low-cost soil gas flux system, we built up on some general guidelines based on experiences from several study sites (Maier et al., 2022). Previous studies developed low-cost solutions for semi-continuous measurements of soil respiration (Gagnon et al., 2016; Forbes et al., 2023).

In the proposed solution, we allow the users to adapt the chamber's dimensions to fit the study site requirements. Of course, the choice of the material of the chamber is flexible. We used here a high light and UV transmission materials to monitor Net 260 Ecosystem Exchange (NEE). Of course, the material needs to be adapted if one is aiming at monitoring only soil respiration, by using opaque materials like PVC. Additionally, in our repository, we made available a open-source software responsible for controlling the systems procedures and the design of all components necessary to integrate the chamber to the sensor. This includes the PCB designs that can easily be modified and fabricated to fit other sensors.

After the chamber design was set, we focused on the sensor capability to measure the evolution of $CO_2$ concentration inside 265 the chamber. Initially, the static test we performed between our proposed choice of low-cost sensor and reference showed a linear correlation in the concentration range $400-800$ ppm, which agrees with other studies that utilized the same sensor (K30) (MacAgga et al., 2024; Gagnon et al., 2016), but we found that this relationship deteriorates around 1000 ppm, at which the low-cost sensor ended up underestimating. For the application to our field site, with a two-day snapshot of measurements, the observed range of $CO_2$ concentrations remained below 1000 ppm, so this limitation did not pose an issue in our case. However, 270 to address scenarios where higher $CO_2$ concentrations might occur, one potential solution would be to increase the chamber height using the parametric design solution. This adjustment would reduce the final $CO_2$ concentration by allowing for a larger chamber volume, thereby mitigating potential sensor limitations at higher concentrations. Another approach is to reduce the

total measurement time at night, which is the simplest option, provided that the measurement duration remains sufficient to ensure an accurate flux calculation.

While most studies evaluated the sensor's static $CO_2$ accuracy (MacAgga et al., 2024; Gagnon et al., 2016; Zawilski and Bustillo, 2024), our experiment allowed us to also analyze the integration of the sensor within the chamber. It evaluated the efficiency of the air mixing and the response lags between the input and measurement from both systems, which could only be compared through dimensional analysis. With our chamber dimensions and fan control settings, we found the chamber design and low-cost sensor to be very satisfactory with similar results compare to our commercial LICOR reference. Although the low-cost sensor has a slower response time than the commercial sensor, both lag times can be used as inputs for soil gas flux calculation as the unmixed phase (deadband), allowing initial data to be filtered out before calculating flux.

In the field, it is expected that the soil gas flux happens mostly via diffusion instead of advection as imposed in the experiment. But for our case, this difference in mechanisms did not influence the results of our sensor/design validation. Bringing our low-cost system to the field for 2-day monitoring showed a snapshot its usage in real case applications with a chamber height of 20 cm and a chamber diameter of 20 cm. We could first monitor the variations of ambient $CO_2$ concentrations by considering the first value reported by the sensor after chamber closing. This minimum ambient values of $CO_2$ concentration during the daytime had similar values as the ones reported by the Atmosphere Thematic Centre (Jungfraujoch) (Emmenegger et al., 2025). We also observed an increase in the initial $CO_2$ concentration during the nighttime. This increase was likely to be linked with the atmosphere's boundary layer's higher stability when compared with daytime, which results in its stratification near the surface. The boundary layer's stratification generally has a higher concentration of $CO_2$ near the surface, which then reduces the diffusion rate through the soil to the atmosphere, thus reducing the soil respiration during the nighttime (Lai et al., 2012; Schneider et al., 2009).

This stratification has possible consequences for the low-cost sensor as it imposes an initial concentration higher during the nighttime compared to the daytime (Figure 7a). This implies that higher concentrations are expected during accumulation during the night, potentially leading toward ranges with higher uncertainties (Figure 5). To prevent these uncertainties, the operators need to consider these values during the design of the chamber and measuring times. Görres et al. (2016) suggested shorter chamber heights ($< 20$ cm) and chamber systems that avoid abrupt movements to maintain the stable atmospheric layer. To account for these disturbances for the low-cost system, one could reduce or completely turn off the fan inside the chamber during the nighttime to avoid those artifacts while increasing the measurement period to compensate for this decrease in air mixing rate, and we could also slow down the closing speed for our chamber to avoid breaking the atmospheric stability.

### 4.3 Final considerations

Finally, this study did not focus on the flux calculation schema for calculating the soil gas flux after the measurement of $CO_2$. Instead we only utilized a non-linear method (H-M method) for its calculation. Other methods, like linear fitting, are simple to apply and can be useful considering that is more precise but less accurate (Venterea et al., 2020). And some progress has been made with ideas of balancing the bias and uncertainty from linear and non-linear methods (Hüppi et al., 2018), along with an understanding of the importance of identifying the optimal time intervals for these flux estimative (Baneschi et al., 2023;

Johannesson et al., 2024). Still, the system design is closely linked with the performance of any particular flux calculation schema. While characterization of analytical uncertainty for soil gas flux (Cowan et al., 2025) is a step forward for determining the theoretical limits of the precision of flux measurements for different chamber systems, it uses the frequentist approach combined with the linear method. Thus, a flux calculation schema that quantifies uncertainty for non-linear models which accounts for sensor's characteristics are still needed to be investigated to better understand low-cost system's limitations.

## 5   Conclusions

The development and implementation of low-cost soil gas flux systems present significant opportunities for advancing the monitoring of hydrological and ecological processes. By enabling high spatiotemporal measurements, these systems open new avenues for disentangling the coupled processes that drive soil gas fluxes, such as the impacts of extreme rainfall events, sudden changes in water table height, and soil management practices like mechanization. It also facilitates the integration into a multi-modal spatial monitoring system, such as with eddy covariance systems that could improve the knowledge of complex settings such as peatlands renaturation goals.

Our approach leverages the flexibility of low-cost sensors and proposes solutions for open design, parametrization and software for operating the system. In addition, we proposed a transient experiment to evaluate and compare the performance of soil gas flux systems in controlled conditions. This methodology enhances quality assurance and helps validate the integration of sensors within the chamber design, offering a comprehensive workflow for system optimization. The proposed design has the potential to significantly expand the monitoring of soil gas fluxes. It brings perspectives for broad applicability that can meet the needs of various applications and field conditions for improved understanding of the dynamics of soil-atmosphere interactions and their responses to environmental changes.

*Code availability.* The code and the parametric chamber design are available in the Github (https://github.com/alexnaoki/SoilGasFlux) (Kobayashi, 2025).

*Author contributions.* AK: conceptualization, instrumentation design, code programming, laboratory experiment, formal analysis, methodology, validation, writing-original draft and editing. CR: conceptualization, formal analysis, methodology, validation, writing-original draft and editing. DK: formal analysis, methodology, writing-review, editing and funding. PB: formal analysis, methodology, conceptualization, writing-review, editing and funding. EM: formal analysis, validation, writing-review and editing. RC: validation, writing-review and editing.

*Competing interests.* The authors declare that they have no known competing financial interests or personal relationships that could have appeared to influence the work reported in this paper.

*Acknowledgements.* We would like to thank Ana Tanaka, Saeed Mhanna, Francesco Scattolini, Laurent Marguet and Roberto Costa for their valuable comments and advice.

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
