# Peer review of "Parametric design for soil gas flux system: a low-cost solution for continuous monitoring"

_EGUsphere, 2025_

## Referee Comment (RC1)

Review:
**Parametric design for soil gas flux system: a low-cost solution for continuous monitoring**
Alex Naoki Asato Kobayashi et al.

*General remarks:*

The paper presents the development of a low-cost, parametric soil gas flux system for continuous monitoring of greenhouse gas (GHG) emissions. The design incorporates a chamber system, integrated sensors, a controller PCB, and software that allows for adaptability to various field conditions. The authors validated the system through laboratory experiments (static and transient CO2 tests) and field tests, highlighting its performance and flexibility. The paper also emphasizes the need for parametrized designs to enhance data quality and spatial coverage in soil gas flux studies.

The paper makes a valuable contribution to the field of environmental monitoring by providing an accessible and flexible solution for soil gas flux measurement. In my opinion, the following suggestions could help improve the paper before publication.

*Abstract:*

The abstract provides a concise overview of the study's objectives, methodology, and significance. It effectively highlights the challenges with current systems and introduces the proposed solution. Nevertheless, the abstract does not mention key results (e.g., sensor accuracy, field deployment outcomes) that would strengthen the impact of the findings.

line 5: prohibits -> prohibit

line 11: demonstrate -> demonstrates

*Introduction:*

The introduction provides a clear context for the challenge of monitoring soil gas flux, reviews existing approaches such as Eddy Covariance and chamber methods, and highlights limitations in current technologies. It effectively conveys the need for a low-cost, adaptable solution.

*M&M:*

The methods section outlines the system's components, the parametric chamber design, and the experimental procedures. However, including additional details would enhance clarity and support the reproducibility of the results:

line 66ff: Section 2.1.1 While the concept of parametric design is clear, the process described in Section 3.1 ("user has to define an initial expected soil gas flux...") is relatively brief in the methods section. More detail here on how the system translates user inputs into recommended dimensions or helps the user navigate the parameter space would enhance clarity.

line 77ff: Section 2.1.2 offers a general overview of the sensors used and the workflow executed on the microcontroller. Please add a bit more details on how the system is set up, powered and data transmission is achieved (if). To better support the manuscript's low-cost approach please also include information on system cost, also in comparison to the used reference device (see below).

line 100ff: $p_s$ in formula (1) is not explained in the text. Additionally, the formula presented in the manuscript is different to the one shown in the python code: mdf = (Aa/(tc*(tc*freq)**(1/2)))*(V*P/(A*R*T)) (see code availability).

line 114ff: I assume the Licor LI-7810 is a reference measurement system, but that's not made clear. The type number appears here for the first time, without mentioning what it is.

line 126: Again Licor LI-7810 is mentioned – here again with another type number: 8200-01S without details given about it.

line 150ff: A map/image of the experimental setup would be a nice addition.

line 160ff: Could you clarify why the non-linear H-M model was chosen?

line 172: Where does the prior knowledge come from?

line 173: setup consistent → setup consists?

***Results:***

line 179: Link is wrong (see code availability)

line 181ff: While Figure 4 provides a useful visualization of the MDF landscape, the text refers to user inputs being used to generate a "list of options." It would be helpful if the Results section—or a referenced appendix—included a brief example of this list or offered more detail on how the system generates and ranks these options based on the input parameters. The reasoning behind the selected dimensions (20 cm height, 20 cm diameter) is well-explained, citing material availability and an acceptable area-to-perimeter ratio. However, this section could be strengthened by presenting the alternative configurations considered and explaining why they were rejected, beyond the limitation of material availability.

figure 4: Units for minimum Flux is missing

line 213ff: While the figure (Figure 6) show the higher lag for the low-cost system compared to the reference, quantifying the difference in response lag (in dimensionless time units) would provide a more concrete measure of this difference and strengthen the comparison. The discussion section later suggests potential modifications to mitigate such lags in the field (slowing chamber closure, turning off fan), which hints at the importance of this observed lag in the lab results.

figure 7: The method states the H-M model was used for flux calculation, but the results do not explicitly mention the variability or uncertainty associated with the calculated fluxes. Given

that the discussion later touches upon different flux calculation schema and their impact on uncertainty, providing some measure of confidence or variability for the calculated fluxes in this section would enhance the result presentation. For example, showing error bars or confidence intervals on the flux data in Figure 7b would be informative.

***Discussion:***

line 264ff: As noted above, quantifying the magnitude of the difference in response lag between the systems would strengthen the comparison here. While the discussion acknowledges the difference, elaborating slightly on its potential practical implications for flux calculation timing could be beneficial, though the later suggestions on mitigating field lags (slowing closure, fan control) do touch upon this implicitly.

line 270: The 2-day monitoring period is rather short to draw the conclusion of the robustness of the design, considering the use of low-cost sensors. Long-term stability of the used sensors and setup should be investigated.

line 287ff: the discussion could briefly elaborate on why selecting the optimal flux calculation schema is particularly important for low-cost systems (due to potentially higher sensor noise or different response characteristics compared to commercial systems) beyond the general statement about the link between design and schema. This would further strengthen the justification for this being a critical area for future work for this specific type of system.

***Code availability:***

DOI/Link is wrong –probably: doi.org/10.5281/zenodo.14748702

***Grammar and spelling:***

quite a few small typos – mainly singular/plural-s and usage of wrong prepositions

---

## Author Comment (AC1)

**Responses to Reviewer #1:**

**Comment 0:** *"… the abstract does not mention key results (e.g., sensor accuracy, field deployment outcomes) that would strengthen the impact of the findings."*
**Reply 0:** Thank you for your suggestion. We included key results in the abstract from our laboratory comparison between our low-cost system and a commercial one: (i) a static experiment demonstrating a strong correlation in $CO_2$ gas concentration up to 500 ppm above background levels, and (ii) transient tests, which showed a comparable response time from both systems.

**Comment 1 & 2:** *"line 5: prohibits -> prohibit"* & *line 11: demonstrate -> demonstrates*
**Reply 1:** Thank you for noticing the mistake.

**Comment 3:** *line 66ff: Section 2.1.1 While the concept of parametric design is clear, the process described in Section 3.1 ("user has to define an initial expected soil gas flux…") is relatively brief in the methods section. More detail here on how the system translates user inputs into recommended dimensions or helps the user navigate the parameter space would enhance clarity.*
**Reply 3:** Navigation within the parameter space is specific to each field. First, it is essential to consider both the range of soil gas fluxes and the sensor characteristics to estimate the ideal $CO_2$ concentration inside the chamber, ensuring it stays above the minimal detectable flux. Additionally, the parametrization is constrained by the recommended area-perimeter ratio for surface representativeness and later limited by the user's material availability. Thank you for your feedback.

**Comment 4:** *line 77ff: Section 2.1.2 offers a general overview of the sensors used and the workflow executed on the microcontroller. Please add a bit more details on how the system is set up, powered and data transmission is achieved (if). To better support the manuscript's low-cost approach please also include information on system cost, also in comparison to the used reference device (see below).*
**Reply 4:** Thank you for your suggestion. Since material costs differ significantly across locations and time for both low-cost and commercial systems, we specified that the cost difference between the sensor (SenseAir K30 FR) and the commercial gas analyzer can be up to a thousand-fold. We are also updating the GitHub repository to include only the estimated cost for the low-cost system.

**Comment 5:** *line 100ff: ps in formula (1) is not explained in the text. Additionally, the formula presented in the manuscript is different to the one shown in the python code: mdf = (Aa/(tc\*(tc\*freq)\*\*(1/2)))\*(V\*P/(A\*R\*T)) (see code availability).*
**Reply 5:** Thank you for your comment. We corrected equation 1 and included a description for ps. Now, both the equation in the text and the referenced Python code are consistent, with equation 1 using seconds [s] and the Python code using frequency [Hz].

**Comment 6:** *line 114ff: I assume the Licor LI-7810 is a reference measurement system, but that's not made clear. The type number appears here for the first time, without mentioning what it is.*
**Reply 6:** You are right, it is the model number. We included additional details to clarify that it is the commercial gas analyzer.

**Comment 7:** *line 126: Again Licor LI-7810 is mentioned – here again with another type number: 8200-01S without details given about it.*
**Reply 7:** Once again, we included additional information to clarify that we were referring to the commercial chamber. Thank you.

**Comment 8:** *line 150ff: A map/image of the experimental setup would be a nice addition.*
**Reply 8:** Thank you for the suggestion. We decided not to include a map or image of the experimental setup because we believe the current schematics adequately convey the essential information for this sensor's paper. However, we will add some photos of the experimental setup to the repository as an example of usage.

**Comment 9:** *line 160ff: Could you clarify why the non-linear H-M model was chosen?*
**Reply 9:** Thank you for your comment. We clarified that our selection of the non-linear approach (H-M model) was because it is considered to be a more accurate than the linear alternative (Baneschi et al., 2023).

Baneschi, I., Raco, B., Magnani, M., Giamberini, M., Lelli, M., Mosca, P., Provenzale, A., Coppo, L., and Guidi, M.: Non-steady-state340 closed dynamic chamber to measure soil $CO_2$ respiration: A protocol to reduce uncertainty, Frontiers in Environmental Science, 10, https://doi.org/10.3389/fenvs.2022.1048948, 2023

**Comment 10:** *line 172: Where does the prior knowledge come from?*
**Reply 10:** For our field experiment, the prior knowledge of the minimal soil respiration rate is based on previous sampling campaigns. We revised the text to clarify how these figures were determined, thank you for your comment.

**Comment 11:** *line 173: setup consistent    setup consists?*
**Reply 11:** Done, thank you.

**Comment 12:** *line 179: Link is wrong (see code availability)*
**Reply 12:** We fixed the link. Thank you.

**Comment 13:** *line 181ff: While Figure 4 provides a useful visualization of the MDF landscape, the text refers to user inputs being used to generate a "list of options." It would be helpful if the Results section— or a referenced appendix—included a brief example of this list or offered more detail on how the system generates and ranks these options based on the input parameters. The reasoning behind the selected dimensions (20 cm height, 20 cm diameter) is well-explained, citing material availability and an acceptable area-to-perimeter ratio. However, this section could be strengthened by presenting the alternative configurations considered and explaining why they were rejected, beyond the limitation of material availability.*
**Reply 13:** In the first part of your comment, we will add a "Usage example" to the Github repository to demonstrate how the choice is made. Concerning the second part about chamber design selection, we expanded beyond material availability and clarified that the maximum expected flux is significant due to the ideal gas concentration range. In this range, the low-cost sensor exhibits less bias compared to the commercial system, which is approximately 150 to 500 ppm above background $CO_2$ levels. Thank you for your feedback.

**Comment 14:** *figure 4: Units for minimum Flux is missing*
**Reply 14:** Thank you, we added the units for the colorbar in Figure 4.

**Comment 15:** *line 213ff: While the figure (Figure 6) show the higher lag for the low-cost system compared to the reference, quantifying the difference in response lag (in dimensionless time units) would provide a more concrete measure of this difference and strengthen the comparison. The discussion section later suggests potential modifications to mitigate such lags in the field (slowing chamber closure, turning off fan), which hints at the importance of this observed lag in the lab results.*

**Reply 15:** We agree that converting the time into seconds offers a clearer and more tangible way to compare the two systems. The commercial system shows a lag of approximately 5 seconds, whereas the low-cost system exhibits a lag of around 20 seconds. Thank you

**Comment 16:** *figure 7: The method states the H-M model was used for flux calculation, but the results do not explicitly mention the variability or uncertainty associated with the calculated fluxes. Given that the discussion later touches upon different flux calculation schema and their impact on uncertainty, providing some measure of confidence or variability for the calculated fluxes in this section would enhance the result presentation. For example, showing error bars or confidence intervals on the flux data in Figure 7b would be informative.*

**Reply 16:** Thank you for your suggestion. We recognize that an additional uncertainty arises from the flux calculation schema used to estimate soil gas flux at each measurement point, and quantifying this remains a research gap. However, analyzing this is beyond our sensor paper's scope. Nonetheless, we included the uncertainty band (16th to 84th percentile) derived from the 60-minute resampling in figure 7b.

**Comment 17:** *line 264ff: As noted above, quantifying the magnitude of the difference in response lag between the systems would strengthen the comparison here. While the discussion acknowledges the difference, elaborating slightly on its potential practical implications for flux calculation timing could be beneficial, though the later suggestions on mitigating field lags (slowing closure, fan control) do touch upon this implicitly.*

**Reply 17:** Thank you. We explained how lag quantification via laboratory experiments can be used as inputs for soil gas flux calculations in both low-cost and commercial systems, based on chamber geometry and fan control. Since the lag differences between systems are not vastly different, the influence of the deadband filtering process remains minimal.

**Comment 18:** *line 270: The 2-day monitoring period is rather short to draw the conclusion of the robustness of the design, considering the use of low-cost sensors. Long-term stability of the used sensors and setup should be investigated.*

**Reply 18:** We agree that a 2-day monitoring period does not prove the robustness of the design; rather, it offers a snapshot of a real case application with specific chamber dimensions. To better demonstrate robustness, a longer period is necessary, but since our method involves parametrizing the chamber geometry, this would also require testing various combinations of chamber heights and diameters over extended monitoring periods. Therefore, we will add to the GitHub repository a longer monitoring period, which we can update over time with different monitoring period amounts and chamber dimensions.

**Comment 19:** *line 287ff: the discussion could briefly elaborate on why selecting the optimal flux calculation schema is particularly important for low-cost systems (due to potentially higher sensor noise or different response characteristics compared to commercial systems) beyond the general statement about the link between design and schema. This would further strengthen the justification for this being a critical area for future work for this specific type of system.*

**Reply 19:** Thank you for your comment. Quantifying uncertainty on a per-measurement basis remains largely unexplored, especially for low-cost systems. Some studies have advanced analytical uncertainty (Cowan et al., 2025), but their approach relies on a frequentist framework and is limited to linear models. We revised Section 4.3 to better address the gap in the flux calculation scheme for soil gas fluxes.

Cowan, N., Levy, P., Tigli, M., Toteva, G., and Drewer, J.: Characterisation of Analytical Uncertainty in Chamber Soil Flux Measurements, European Journal of Soil Science, 76, e70 104, https://doi.org/10.1111/ejss.70104, 2025.370

**Comment 20:** *DOI/Link is wrong –probably: doi.org/10.5281/zenodo.14748702*
**Reply 20:** Thank you for noticing. We updated the link in the text.

---

## Author Comment (AC2)

**Responses to Reviewer #2:**

**Comment 1:** *It is not instantly clear how the transient change in concentration in the chamber was actually measured. In Eq. (2), what are the measured variables?*

**Reply 1:** Thank you for your comment. In the transient experiment, after closing the chamber, a standard gas bag is injected inside, gradually increasing the $CO_2$ concentration within the chamber. Both low-cost and commercial gas analyzers measure this increase over time. Therefore, the change in $CO_2$ concentration is determined by the difference in its levels over time. In equation 2, the term refers to the modeled $CO_2$ gas concentration in the chamber from this experiment, expressed in terms of $CO_2$ gas mass rather than concentration.

**Comment 2:** *Figure 7. I don't understand why D[CO2] is shown here instead of the actual [CO2] with Y-axis starting from e.g. 400ppm?*

**Reply 2:** In figure 7a, we used $\Delta CO_2$ instead of the actual $CO_2$ levels, considering the atmospheric $CO_2$ fluctuations from the Atmosphere Thematic Centre (Jungfraujoch) rather than an arbitrary $CO_2$ concentration. Thank you for your question.

**Comment 3:** *258-261: Or the measurement length could be shortened.*

**Reply 3:** We agree that an alternative is to change the measurement length. However, it is important to ensure the measurement duration remains long enough for accurate flux calculation. Thank you for your suggestion; we will include this in the text.

**Comment 4:** *283-285: Another option would be to introduce constant air mixing. As long se the conditions do not change, there should not be gas accumulation/release (storage change) effects.*

**Reply 4:** Our current low-cost design depends on the lid of the opening and closing chamber to initiate the measurement. Therefore, we believe that the option of constant air mixing, which is ideal for a static chamber, is not suitable for our setup. Thank you for your suggestion.

**Comment 5:** *290: of a any -> remove "a"*

**Reply 5:** Thank you.

**Comment 6**: *297: Please specify here what you mean by high-frequency. Spatial, temporal or spatiotemporal?*

**Reply 6:** We referred to high spatiotemporal measurements because they enable automatic measurements without user interaction. Their low cost allows for increased spatial coverage and scalability of monitoring capacity. Thank you.

**Comment 7:** *309: The link seems to be wrong here. It is possible to navigate to the correct page (https://github.com/alexnaoki/SoilGasFlux; I assume) from the link in the references, though. Opening*

*the chamber schematic requires an OnShape account that I did not want to create, and I will not comment on that.*

**Reply 7***:* We fixed the link and updated the GitHub repository.